# Pharmacological Activities of Soursop (*Annona muricata* Lin.)

**DOI:** 10.3390/molecules27041201

**Published:** 2022-02-10

**Authors:** Mutakin Mutakin, Rizky Fauziati, Fahrina Nur Fadhilah, Ade Zuhrotun, Riezki Amalia, Yuni Elsa Hadisaputri

**Affiliations:** 1Department of Pharmaceutical Analysis and Medicinal Chemistry, Faculty of Pharmacy, Universitas Padjadjaran, Bandung 45363, Indonesia; 2Department of Pharmaceutical Biology, Faculty of Pharmacy, Universitas Padjadjaran, Bandung 45363, Indonesia; rizky17007@mail.unpad.ac.id (R.F.); fahrina17001@mail.unpad.ac.id (F.N.F.); ade.zuhrotun@unpad.ac.id (A.Z.); 3Department of Pharmacology and Clinical Pharmacy, Universitas Padjadjaran, Bandung 45363, Indonesia; riezki.amalia@unpad.ac.id

**Keywords:** soursop, *Annona muricata* L., traditional medicine, pharmacological activities

## Abstract

Soursop (*Annona muricata* Lin.) is a plant belonging to the Annonaceae family that has been widely used globally as a traditional medicine for many diseases. In this review, we discuss the traditional use, chemical content, and pharmacological activities of *A.muricata*. From 49 research articles that were obtained from 1981 to 2021, *A.muricata*’s activities were shown to include anticancer (25%), antiulcer (17%), antidiabetic (14%), antiprotozoal (10%), antidiarrhea (8%), antibacterial (8%), antiviral (8%), antihypertensive (6%), and wound healing (4%). Several biological activities and the general mechanisms underlying the effects of *A.muricata* have been tested both in vitro and in vivo. *A.muricata* contains chemicals such as acetogenins (annomuricins and annonacin), alkaloids (coreximine and reticuline), flavonoids (quercetin), and vitamins, which are predicted to be responsible for the biological activity of *A.*
*muricata*.

## 1. Introduction

Presently, the use of natural ingredients as treatments for various diseases is increasing. Plants are a source of natural ingredients that are widely used as medicines. The compounds present in plants are responsible for their activities against various diseases, and studies can be performed to identify the active compounds in plants and determine their pharmacological activities against diseases [1]. 

Many studies on plants, their contents, and the pharmacological activities of their constituents have been conducted. *Annona muricata* Lin., commonly called soursop, is part of the Annonaceae family, which comprises more than 130 genera and 2300 species. *A. muricata* L. contains various compounds with pharmacological activity. This plant is widely grown in tropical and subtropical areas, such as Southeast Asia, South America, and the rainforests of Africa. The plant produces edible fruit all year round and is widely used as a traditional medicine for skin disease, respiratory disease, fever, bacterial infections, diabetes, hypertension, and cancer [1,2]. Different parts of *A.muricata* have different activities. The seeds combat parasitic infections; the fruit is used for the treatment of arthritis, nervous disorders, and diarrhea; and the leaves are used to treat cystitis, headaches, insomnia, and cancer [3].

The main active components of *A. muricata* are acetogenin, alkaloids, and flavonoids [4]. Analysis of the compounds in *A. muricata* leaf extract revealed secondary metabolites such as flavonoids, terpenoids, saponins, coumarins, lactones, anthraquinones, glycosides, tannins, and phytosterols [5].

In this review, we provide an overview of the botanical description, traditional uses, compounds, and pharmacological activities of *A.muricata*. We summarized the existing literature relevant to *A. muricata*, its compound contents, pharmacological activities, and the mechanisms of its pharmacological activity published during the period of 1981–2021. 

## 2. Botanical Description

*A. muricata*, known as guanabana, soursop, graviola, or Brazilian paw paw [1,2], is a native plant of Central America [3]. This plant is distributed widely throughout Southeast Asia, South America, and the rainforests of Africa [1]. *A.muricata* is commonly known as soursop because of the sweet and sour taste of its fruit. In Portuguese, *A.muricata* is known as graviola; in Latin America, it is known as guanabana; and in Indonesia, it is known as nangka belanda or sirsak. Other traditional names include annone, araticum, araticum-manso, anona, anoda, coronsol, grande, grand corossol, gurusulu, quanabana, sauersack, taggannona, and zuurzak. *A.muricata* is a fruit-bearing plant that belongs to the kingdom Plantae, the division Angiospermae (Magnoliophyta), the class Magnolid, the order Magnoliales, the family Annonaceae family, and the genus *Annona* [6,7].

The *A.muricata* tree grows at altitudes below 1200 m above sea level, at a relative humidity of 60%–80%, a temperature ranges of 25–28 °C, and with more than 1500 mm of annual rainfall [6,7]. *A.muricata* is an evergreen plant that blooms and bears fruit almost throughout the year [6]. The leaves are obovate, oblate, and acuminate, with a dark green, thick, and glossy upper surface. Figure 1 shows the *A.muricata* tree, leaves, fruit, and flowers. The fruit is green and heart-shaped, with soft prickly skin containing juicy, aromatic, and acidic pulp [1,4].

*A.muricata* has been widely used to treat many disorders, such as parasitic infections, inflammation, diabetes, and cancer [8]. All parts of *A.muricata* are used in traditional medicine by people who live in tropical areas, with the leaves, stem bark, roots, and seeds primarily used as medicinal ingredients [9]. *A.muricata* leaves are used to treat headaches, insomnia, cystitis, and cancer, the seeds are used to treat parasitic infections [1], and the fruit is used to treat diarrhea and neuralgia, eliminate worms and parasites, increase milk production in lactating women, and reduce fever [10].

## 3. Traditional Uses

Ethnobotanical studies have reported that *A.muricata* is used to treat bacterial and fungal infections, as it possesses anthelmintic, antihypertensive, anti-inflammatory, and anticancer activities. It has also been used as an analgesic and to treat fever, respiratory and skin illnesses, diabetes, and internal and external parasites. In several tropical sub-Saharan countries such as Uganda, all parts of the plant are used to treat malaria, stomach ache, parasitic infections, diabetes, and cancer [7,11].

Additionally, the seeds are used as anthelmintic and antiparasitic treatments, and the leaves, bark, and roots of *A.muricata* have been used for their anti-inflammatory, antihypertensive, sedative, antidiabetic, smooth muscle relaxant, and antispasmodic effects [1,12,13]. The leaves are used to treat cystitis, diabetes, headaches, hypertension, insomnia, and liver problems and as an antidysenteric, anti-inflammatory, and antispasmodic agent. The cooked leaves are applied topically to treat abscesses [14]. In tropical African countries, including Nigeria, the leaves are traditionally used to treat skin diseases [12].

In South America, *A.muricata* fruit juice is used to treat many diseases, such as heart and liver disease, and has antidiarrhea and antiparasitic effects [15]. The fruit flesh is used to increase breast milk production after childbirth and treat rheumatism, arthritic pain, fever, neuralgia, dysentery, heart and liver diseases, and skin rashes, and it has antidiarrhea, antimalarial, antiparasitic, and anthelmintic properties [1,16]. Table 1 summarizes the results of previous studies on the pharmacological activities of *A.muricata* and the underlying molecular mechanisms.

**Table 1 molecules-27-01201-t001:** Pharmacological activities of *A. muricata*.

Pharmacological Activity	Plant Parts	Mechanisms	Ref.
Anticancer	Fruit, stem, seed, and twigs	Inhibits MMP-2 and MMP-9, which play an important role in cancer progression, in HT1080 fibrosarcoma cells.	[17]
	Leaf, twigs, and root	Disrupts MMP function, reactive oxygen species (ROS) generation, and G0/G1 cell cycle arrest in HL-60 leukemia cells.	[10]
	Leaf	Increases Bax expression and decreases Bcl-2 expression, cell cycle arrest at G0/G1 phase in A-549 lung cancer cells.	[18]
		Induces apoptosis by enhancing caspase-3 expression in COLO-205 colorectal cancer cells.	[19]
		Induces apoptosis by enhancing the expression of caspase-3 in MDA-MB-231 breast cancer cells.	[20]
		Inhibits the proliferation of PC-3 human prostate cancer cells.	[21]
		Disrupts MMP function, causes leakage of cytochrome C from mitochondria, and activates caspase-3, caspase-7, and caspase-9 expression in HT-29 colon cancer cells.	[22]
		Apoptosis mechanism mediated by a decrease in Bcl-2 expression and an increase in caspase-3 and caspase-9 expression in MCF7 breast cancer cells.	[23]
	Seeds	Increases caspase-3 cleavage and DNA fragmentation in endometrial cancer cells.	[24]
Antiulcer	Leaf	Activates prostaglandin synthesis and supresses aggressive factors of gastric mucosa.	[25]
		Protects against ROS scavenging and gastric wall damage.Upregulates Hsp70 and downregulates Bax expression.	[26]
		Downregulates Bax and malondialdehyde (MDA) expression.Upregulates CAT, SOD, GSH, NO, PGE2, glycogen, and Hsp70 expression.	[27]
Antidiarrhea	Fruit	Inhibits intestinal motility and secretions.	[28]
Antiprotozoal	Leaf	Antiprotozoal activity against *Toxoplasma gondii*.	[29]
		Antiprotozoal activity against *Leishmania* spp. and *Trypanosoma cruzii*.	[30]
		Antiprotozoal activity against *Plasmodium falciparum*.	[29]
	Seeds	Antiprotozoal activity against *Leishmania* spp.	[31]
	Bark and roots	Antiprotozoal activity against *Plasmodium falciparum*.	[32]
Antidiabetic	Fruit	Inhibits α-amylase and α-glucosidase enzymes.	[33]
	Leaf	Decreases lipid peroxidation and indirectly affects insulin production and endogenous antioxidants in streptozotocin-induced mice.	[12]
Antibacterial	Leaf	Attacks the bacterial membrane.	[34]
Antihypertensive	Fruit and leaves	Inhibits angiotensin-I-converting enzyme and blocks calcium ion channels	[33,35]

## 4. Phytochemical Properties

Various compounds and secondary metabolites are present in the *A.muricata* plant (Table 2). The major compounds are acetogenins, alkaloids, flavonoids, essential oils, vitamins, carotenoids, amides, and cyclopeptides [4,36]. Additionally, the plant contains minerals such as K, Ca, Na, Cu, Fe, and Mg [37].

Among the major compounds, acetogenin is the most abundant in *A.muricata*. Acetogenin is a long-chain fatty acid derivative that is widely present in the Annonaceae family and is produced via the polyketide pathway. Acetogenins have a long aliphatic chain of 35–38 carbons bonded to a g-lactone a-ring, terminally substituted by b-unsaturated methyl, with tetrahydrofurans (THF) located along the hydrocarbon chain (Figure 2) [38,39]. The most abundant alkaloid compounds in *A.muricata* are reticuline and coreximine (Figure 3).

The most common alkaloids present in this plant are the isoquinoline, aporphine, and protoberberine types [4,40]. The most common flavonoid is quercetin (Figure 4) [41,42,43], although the most abundant flavonoid in the leaf extract is rutin, followed by quercetin and kaempferol [21].

**Table 2 molecules-27-01201-t002:** Phytochemical properties of *A. muricata*.

No.	Compound	Part of Plant	Type	Refs.
1	Anomuricine	Leaf, root, stem, bark	Alkaloid	[44]
2	Anomurine	Leaf, root, stem, bark	Alkaloid	[44]
3	Annonaine	Fruit, leaf	Alkaloid	[45,46]
4	Annonamine	Leaf	Alkaloid	[47]
5	Asimilobine	Fruit	Alkaloid	[45,46,48]
6	Atherospermine	Stem	Alkaloid	[44]
7	Atherosperminine	Root, bark	Alkaloid	[44]
8	Casuarine	Leaf	Alkaloid	[40]
9	Coclaurine	Root, bark	Alkaloid	[44,45]
10	Coreximine	Leaf, root, stem, bark	Alkaloid	[44]
11	DMDP (2,5- Dihydroxymethyl-3,4, dihydroxypyrrolidine)	Leaf	Alkaloid	[40]
12	deoxymannojirimycin	Leaf	Alkaloid	[40]
13	deoxynojirmycin	Leaf	Alkaloid	[40]
14	(*R*)-O,O-dimethylcoclaurine	Leaf	Alkaloid	[47]
15	Isoboldine	Leaf	Alkaloid	[45]
16	Isolaureline	Leaf	Alkaloid	[48]
17	Liriodenine	Leaf	Alkaloid	[45]
18	(*R*)-4’O-methylcocaurine	Leaf	Alkaloid	[47]
19	*N*-methylcoclaurine	Leaf	Alkaloid	[45]
20	(*S*)-narcorydine	Leaf	Alkaloid	[47]
21	Nornuciferine	Fruit	Alkaloid	[46]
22	Remerine	Leaf	Alkaloid	[45]
23	Reticuline	Leaf, root, stem, bark	Alkaloid	[44]
24	Stepharine	Leaf	Alkaloid	[44]
25	Swainsonine	Leaf	Alkaloid	[40]
26	Xylopine	Leaf	Alkaloid	[48]
27	15-acetylguanacone	Fruit	Acetogenin	[49]
28	Annocatalin	Leaf	Acetogenin	[50]
29	Annocatacin A	Seed	Acetogenin	[51]
30	Annocatacin B	Leaf	Acetogenin	[51]
31	Annomontacin	Seed	Acetogenin	[50]
32	Annomuricin	Leaf	Acetogenin	[52]
33	Annomuricin A	Leaf	Acetogenin	[53]
34	Annomuricin B	Leaf	Acetogenin	[53]
35	Annomutacin	Leaf	Acetogenin	[54]
36	Annonacin	Leaf, seed	Acetogenin	[50,54]
37	Annonacin A	Leaf	Acetogenin	[54]
38	Annonacinone	Leaf	Acetogenin	[50]
39	Annoreticuin-9-one	Seed	Acetogenin	[55]
40	Annoreticuin, cis	Pulp	Acetogenin	[55]
41	Bullatacin	Seed	Acetogenin	[38]
42	Cohibin A	Root	Acetogenin	[56]
43	Cohibin B	Seed	Acetogenin	[56]
44	Cohibin C	Seed	Acetogenin	[57]
45	Cohibin D	Seed	Acetogenin	[57]
46	Corepoxylone	Seed	Acetogenin	[58]
47	Corossolin	Seed, leaf	Acetogenin	[59,60]
48	Corossolone	Leaf	Acetogenin	[50]
49	Epomurinins A, B	Pulp	Acetogenin	[61]
50	Epomurisenins A, B	Pulp	Acetogenin	[61]
51	Gigantecin	Seed, leaf	Acetogenin	[60]
52	2,4 Cis or trans Gigantetrocinone	Seed	Acetogenin	[62]
53	Gigantetronenin	Leaf, seed	Acetogenin	[63]
54	Goniothalamicin	Seed, leaf	Acetogenin	[64]
55	Javoricin	Seed	Acetogenin	[64]
56	Longifolicin	Seed	Acetogenin	[59]
57	Montanacin	Leaf	Acetogenin	[60]
58	Montanacin D	Leaf, pulp	Acetogenin	[60]
59	Montanacin E	Leaf, pulp	Acetogenin	[60]
60	Montanacin H	Leaf	Acetogenin	[60]
61	Montecristin	Pulp	Acetogenin	[60]
62	Muricatenol	Seed	Acetogenin	[65]
62	Muricatetrocin A	Seed	Acetogenin	[59]
63	Muricatetrocin B	Seed	Acetogenin	[59]
64	Muricatocin A	Leaf	Acetogenin	[66]
65	Muricatocin B	Leaf	Acetogenin	[66]
66	Muricatocin C	Leaf	Acetogenin	[63]
67	Muricin	Seed	Acetogenin	[50]
68	Muricenin	Pulp	Acetogenin	[67]
69	Murisolin	Seed	Acetogenin	[50]
70	Sabadelin	Pulp	Acetogenin	[55]
72	Solamin	Leaf	Acetogenin	[50]
73	Xylomaticin	Seed	Acetogenin	[50]
74	Apigenin-6-C-glucoside	Leaf	Flavonoid	[68]
75	Argentinine	Leaf	Flavonoid	[41]
76	Catechin	Leaf	Flavonoid	[41]
77	Coumaric acid	Pulp	Flavonoid	[69]
78	Daidzein	Leaf	Flavonoid	[68]
79	Dihydrokaempferol-hexoside	Pulp	Flavonoid	[69]
80	Epicatechin	Leaf	Flavonoid	[41]
81	Gallocatechin	Leaf	Flavonoid	[68]
82	Genistein	Leaf	Flavonoid	[68]
83	Glycitein	Leaf	Flavonoid	[68]
84	Homoorientin	Leaf	Flavonoid	[68]
85	Isoferulic acid	Leaf	Flavonoid	[68]
86	Kaempferol	Leaf,	Flavonoid	[41]
87	Quercetin	Leaf	Flavonoid	[41]
88	Quercetin-3-*O*-glucoside	Leaf	Flavonoid	[41]
89	Robinetin	Leaf	Flavonoid	[68]
90	Tangeretin	Leaf	Flavonoid	[68]
91	Rutin	Leaf	Flavonoid	[21]
92	Gallic acid	Leaf	Tannin	[41,68]
93	Vitamin C	Pulp, leaf	Vitamin	[36]
94	Vitamin E	Leaf, seed, pulp	Vitamin	[36]
95	Annoionol A	Leaf	Megastigmane	[70]
96	Annoionol B	Leaf	Megastigmane	[70]
97	Annoionol C	Leaf	Megastigmane	[70]
98	Annoionoside	Leaf	Megastigmane	[70]
99	*N*-p-coumaroyl tyramine	Leaf	Amide	[54]
100	Annomuricatin A	Seed	Cyclopeptides	[71]
101	Annomuricatin B	Seed	Cyclopeptides	[72]
102	Annomuricatin C	Seed	Cyclopeptides	[3]

## 5. Pharmacological Activities

### 5.1. Anticancer

The anticancer activity of *A.muricata* is related to its cytotoxic activity against cancer cells. Table 3 shows the effects of *A.muricata* against cancer cells.

Extracts from several parts of *A.muricata* act as anticancer agents via several mechanisms. Reportedly, the extracts of the fruit, stems, seeds, and twigs of *A.muricata* administered to fibrosarcoma cells (HT1080) inhibited matrix metalloproteinases (MMPs) such as MMP-2 and MMP-9, which play important roles in cancer progression [17]. Extracts from the leaves, twigs, and roots inhibited the proliferation of the human leukemia cell line HL-60 by disrupting MMPs, reactive oxygen species (ROS) generation, and the G0/G1 cell cycle arrest that led to the inhibition of cancer cell growth [10] (Figure 5). Treatment of the A549 lung cancer cell line with the ethyl acetate extract of *A.muricata* leaves induced apoptosis via the upregulation of Bax and downregulation of Bcl-2 expressions. It has also been reported that apoptosis induced by the ethyl acetate extract of *A.muricata* leaf is related to cell cycle arrest at the G0/G1 phase [18]. The leaf extract induced apoptosis by enhancing the expression of caspase-3 in the colorectal cancer cell line COLO-205 [19] and the breast cancer cell line MDA-MB-231 [20]. Another study showed that the ethanol extract and ethyl acetate fractions of *A.muricata* leaves were active against MCF7 cells via an apoptosis mechanism mediated by decreased Bcl-2 expression and increased caspase-3 and caspase-9 expression [23]. Additionally, the presence of annonaceous acetogenins along with flavonoids in *A.muricata* leaves inhibited the proliferation of the human prostate cancer cell line PC-3 [21]. Several compounds isolated from *A.muricata* also show antiproliferative effects. Annomuricin E inhibited HT-29 cell growth by disrupting MMPs, causing leakage of cytochrome c from mitochondria, and activating the pro-apoptotic factors caspase-3, caspase-7, and caspase-9 [22]. Annonacin inhibited the proliferation of endometrial cancer cell lines ECC-1 and HEC-1A via annonacin-mediated apoptotic cell death, which was associated with an increase in caspase-3 cleavage and DNA fragmentation [24]. Another mechanism related to the anticancer activity of *A. muricata* is the modulation of antioxidant enzyme activities. A study reported that 50% ethanol extract of *A. muricata* leaves led to the upregulation of the expression of the antioxidant enzyme superoxide dismutase-1 (SOD1), which catalyzes the breakdown of superoxide into oxygen (O_2_) and hydrogen peroxide (H_2_O_2_), preventing cellular damage [73].

A study that administered 300 mg of *A. muricata* leaf water extract to patients with colorectal cancer in capsule form after breakfast reported the inhibition of colorectal cancer cell growth (DLD-1 and COLO 205). The *A.*
*muricata* leaf water extract has selective inhibitory activity against colorectal cancer cells and does not inhibit normal cell growth. The inhibition of cancer cell growth is modulated by acetogenin activity in the complex I mitochondrial electron transport chain, hampering the process of ATP formation needed for cancer cell growth [22] (Figure 5). Other studies also showed the inhibitory effects of acetogenin against colon cancer cells. Consumption of 5 g of leaf extract powder and seeds of *A. muricata* three times per day accompanied by lifestyle modifications was shown to help the healing process in patients with colon cancer. Another study showed that one of the acetogenins in *A. muricata*, annocherimolin, has cytotoxic potential against HT-29 colon cancer cells [74].

**Table 3 molecules-27-01201-t003:** IC_50_ of several parts of *A. muricata* against cancer cell lines.

Plant Parts	Cancer Cell Line	IC_50_	Refs.
Leaf, twig, and root	HL-60 human leukemia cell line	6–49 µg/mL	[10]
Leaf	A-549 lung cancer cell line	5.09 µg/mL	[18]
Leaf	MCF7 breast cancer cell line	Ethanolic extract: 5.3 µg/mL; ethyl acetate fraction: 2.86 µg/mL; *n-*hexane fraction: 3.08 µg/mL; water fraction: 48.31 µg/mL	[23]
Leaf	PC-3 human prostate cancer cell line	63 µg/mL	[21]
Annomuricin E from leaves	HT-29 colon cancer cell line	1.62 µg/mL	[22]
Annoniacin from seeds	ECC-1 and HEC-1 human endometrial cancer cell lines	4.62–4.75 µg/mL	[24]

### 5.2. Antiulcer

*A.muricata* contains a high concentration of flavonoids, tannins, and phenolic acids, which possess therapeutic effects due to their antioxidant, anti-inflammatory, and gastroprotective properties [25,75,76]. A survey revealed that the leaves and bark of *A.muricata* are popularly used to make tea to treat gastrointestinal problems such as gastritis and poor digestion [25].

In several studies, *A.muricata* has been reported to improve gastric lesions. A study that used a hydroalcoholic extract of *A.muricata* leaves to treat ulcers in absolute or acidified methanol- or indomethacin-induced gastric lesions in rats showed that the extract reduced the ulceration process by activating prostaglandin synthesis as a gastro protector and suppressed aggressive factors of the gastric mucosa [25]. In another study, *A.muricata* ethyl acetate extract showed antiulcer activity via ROS-scavenging and gastric wall damage protection in rats with ethanol-induced gastric injury. Additional mechanisms of *A.muricata* antiulcer activity include the upregulation of Hsp70 and the downregulation of Bax, which are involved in gastric injury suppression [26]. The minimal inhibitory concentration of *A.muricata* leaf extract against *H*. *pylori* is 20 mg/mL [77]. *A.muricata* also showed antiulcer activity by downregulating the expressions of Bax and MDA and upregulating the expressions of catalase (CAT), superoxide dismutase (SOD), glutathione (GSH), nitric oxide (NO), PGE2, glycogen, and Hsp70 [27]. *A.muricata* also inhibited inflammatory mediators such as IL-1β, TNF-a, and IL-6 [78] (Figure 6).

### 5.3. Antidiarrhea

Diarrhea is a common gastrointestinal disorder caused by bacterial infections. It is characterized by abdominal pains, watery stools, and increased bowel movement frequency [79]. Antibiotics have been used as antidiarrhea drugs; however, disadvantages such as bacterial resistance and adverse side effects limit their usefulness [80]. Traditional plants are also widely used to treat diarrhea. The bark and fruit of *A.muricata* are widely used by West Africans to treat diarrhea, and their antidiarrhea effects have been reported [81,82]. The fruit of *A.muricata* showed antidiarrhea activity at a dose of 400 mg/kg body weight in mice with castor oil-induced diarrhea. Flavonoids, triterpenoids, and saponins of *A.muricata* play a role in its antidiarrhea activity by inhibiting intestinal motility and secretions that cause diarrhea [28].

### 5.4. Antidiabetic

*A.muricata* also exhibits antidiabetic activity. It contains flavonoids that inhibit α-glucosidase activity through hydroxylation bonding and substitution at the b-ring. This inhibition suppresses carbohydrate hydrolysis and glucose absorption and inhibits carbohydrate metabolism into glucose [83].

*A.muricata* fruit extracts were reported to exert antioxidant and antidiabetic effects by inhibiting key enzymes relevant to type 2 diabetes mellitus, such as α-amylase and α-glucosidase, in vitro. A study showed that the pericarp of *A.muricata* has the highest inhibitory enzyme and antioxidant properties [33]. The fruit pulp and leaf extract also showed high abilities to inhibit α-amylase and α-glucosidase and minimize the rate of glucose assimilation into the blood after feeding compared with the standard drug [84]. The aqueous extract of *A. muricata* shows antidiabetic effects via antioxidant mechanisms. *A. muricata* leaf extract given to streptozotocin-induced diabetic mice induced a decrease in lipid peroxidation processes, which are a sign of oxidative stress, and indirectly affected insulin production and endogenous antioxidants [12].

*A. muricata* seed oil also showed potential antidiabetic activity against type 1 diabetes induced by streptozotocin. The study showed that an experimental model treated with *A. muricata* seed oil had significantly reduced blood glucose levels compared with the control group. The preserved area of the pancreatic islets was also improved compared with that in the control group [85]. Another study indicated that diabetic rats treated with *A.muricata* had significantly reduced blood glucose levels. That study also reported that daily intraperitoneal administration of 100 mg/kg *A.muricata* extracts to diabetic rats for 15 consecutive days resulted in a statistically significant increase in body weight despite a decrease in food and fluid intake, which is an indicator of improved glycemic control [86].

### 5.5. Antiprotozoal

Diseases caused by parasitic protozoa such as toxoplasmosis, trypanosomiasis, leishmaniasis, and malaria are the most common protozoal diseases worldwide [30]. The lack of treatment options for parasitic protozoal infections implies the importance of developing therapies for protozoal diseases by utilizing the potential of medicinal plants.

Several studies have been conducted to determine the antiprotozoal activity of *A.muricata*. A study reported that *A.muricata* ethanol leaf extract showed antiprotozoal activity against *Toxoplasma gondii* with an IC_50_ of 113.3 µg/mL [29]. Another study reported that *A.muricata* ethyl acetate leaf extract showed antiprotozoal activity against *Leishmania* spp. and *Trypanosoma cruzii* with IC_50_s of < 2 5 µg/mL [30]. Moreover, some studies detected antiprotozoal activities in several compounds isolated from *A.muricata*. Two acetogenins, annonacinone and corossolone, that were isolated from the seeds of *A.muricata* showed antileishmanial activity, with an IC_50_ ranging from 13.5 to 37.6 µg/mL [87]. *A.muricata* ethanol leaf extract also has antiprotozoal activity against *Plasmodium falciparum* with an IC_50_ 46.1 µg/mL [29]. Additionally, the dichloromethane fractions and subfractions of *A.muricata* bark and roots showed antiprotozoal activity against *P. falciparum*, with IC_50_ values ranging from 0.07 to 3.46 µg/mL. Furthermore, gallic acid compounds isolated from the bark and roots of *A.muricata* showed activity against *P. falciparum* with an IC_50_ 3.32 µg/mL [32].

### 5.6. Antibacterial

*A.muricata* extracts showed antibacterial activity against Gram-positive and Gram-negative bacteria compared with the standard antibiotic streptomycin. However, the solvent used for extraction can affect the bioactive efficacy of the extracts. The combination of *A.muricata* ethanolic extract and antibiotic treatment decreased the potential of antibiotic multidrug-resistant *Escherichia coli* and *Staphylococcus aureus* strains [88,89,90]. Another study reported that the bioactive compounds in *A. muricata*, such as alkaloids (annonaine, asimilobine, liriodenirine, nornuciferine, etc.), attack the bacterial membrane (plasma and outer membrane), resulting in broad-spectrum antibacterial activity [34].

### 5.7. Antiviral

*A.muricata* extract has been reported to possess antiviral activity, for example by interfering with the replication process of HIV-I. In another study, ethanolic extracts from the stem and bark of *A.muricata* showed in vitro antiviral activity against herpes simplex virus [91,92]. Another study showed that acidified ethanolic extract of *A.muricata* decreased viral replication after 1 h of contact time. This activity may be due to the presence of phenolic compounds such as rutin [93]. Acetogenins such as annomuricin a, annomuricin b, annomuricin c, muricatocin c, muricatacin, cis-annonacin, annonacin-10-one, cis-goniothalamicin, arianacin, and javoricin were shown to possess good inhibitory activity against SARS-CoV-2 spike proteins (in silico). Cis-annonacin had a low binding energy and greater hydrogen bond formation ability, which indicated that it was the most potent among the acetogenins tested in the study. This result shows that annonaceous acetogenins can be viewed as potential anti-SARS-CoV-2 agents and should be studied in vitro and in vivo [94].

### 5.8. Antihypertensive

Research has revealed that *A.muricata* fruit extracts exhibit antioxidant and antihypertensive properties through angiotensin-I-converting enzymes in vitro [33]. *A.muricata* leaf extract was also shown to show antihypertensive activity in normotensive rats. The suggested hypotensive mechanism of action of *A.muricata* extract is via the blockage of calcium ion channels, which lowers blood pressure [35]. Another study showed that *A. muricata* aqueous extract had antihypertensive properties; combinations of *A. muricata* and other plants, such as *Persea americana*, also exhibited antihypertensive activity, providing a safe and effective solution for hypertension prevention and treatment [95].

### 5.9. Wound Healing

*A. muricata* is also known for its wound healing activity. Two doses of *A. muricata* ethyl acetate extract showed significant wound healing activity in both macroscopic and microscopic analyses of wounds. Wound treatment with an ointment containing *A. muricata* ethyl acetate extract caused a significant increase in antioxidant levels and a decrease in the MDA level in wound tissues compared with those in the vehicle control [96]. *A. muricata* bark and leaf extracts also showed wound healing effects compared with that in untreated wounds [97].

## 6. Toxicology

Several studies have been conducted to determine the toxicity of *A. muricata*. The level of toxicity of *A. muricata* depends on the part of the plant as well as the solvent. A study showed that *A. muricata* aqueous extract had an LD_50_ > 5 g/kg, whereas that of the ethanolic extract was >2 g/kg [14]. Another study reported an LD_50_ of >211 mg/kg for *A. muricata* leaf aqueous extract, which is higher than the recommended daily consumption limit for humans. The aqueous extract of *A. muricata* with doses of >1 g/kg can cause hypoglycemic conditions and hyperlipidemia, and doses of >5 g/kg can cause damage to the kidneys [98].

A study reported that acetogenin in *A. muricata* is a neurotoxin that has the potential to cause neurodegenerative disorders [99]. Acetogenin causes an increase in tau phosphorylation, which is associated with neurodegenerative tauopathy [100]. In addition, some alkaloids in *A. muricata* are believed to have an influence on nerve cells [1]. Annonacin, the most abundant acetogenin in *A. muricata*, as well as some types of alkaloids, such as reticuline, solamin, and coreximine, disrupt the energy formation process in dopaminergic cells [4]. In murine model tests, annonacin was demonstrated to penetrate the brain barrier, decrease ATP levels in brain cells, and damage the basal ganglia [101]. In mice, annonacin caused a decrease in ATP levels in the striatum and disrupted energy production by mitochondria, resulting in the disruption of tau cells, which led to symptoms of neurodegenerative disease [99].

Although some compounds in *A. muricata* have been reported to play a role in neurodegenerative disorders, the doses that produced negative effects were equivalent to consuming one fruit every day for 1 year [102]. Research on the neurotoxicity of annonacin showed that neurodegenerative conditions caused by these compounds arise due to continuous exposure or consumption. Thus, to avoid the occurrence of neurodegenerative conditions that may occur due to compounds present in *A. muricata*, continuous excessive consumption is not recommended [103].

## 7. Conclusions

*A.muricata* is widely used as a traditional medicine. Parts of the *A.muricata* plant, such as the leaves, fruit, seeds, bark, and roots, have pharmacological properties. From the 49 research articles that we obtained, it was reported that its pharmacological properties included anticancer (25%), antiulcer (17%), antidiabetic (14%), antiprotozoal (10%), antidiarrhea (8%), antibacterial (8%), antiviral (8%), antihypertensive (6%), and wound healing properties (4%) (Figure 7), because of the various compounds contained in *A. muricata*. Meanwhile, from 35 reference articles, 101 single compounds of *A. muricata* were reported. The main active compounds in *A. muricata* are acetogenins (49%), alkaloids (26%), flavonoids (19%), and others (6%), which are reported to be responsible for the pharmacological activities listed above (Figure 8). However, not all secondary metabolites of *A. muricata* have been identified.

Many studies on *A. muricata* were conducted in the past three decades; however, no preparations produced from *A. muricata* have been tested and approved by the FDA or EMA. Acetogenins, which are the main active compounds, are difficult to obtain because they are thermolabile, creating challenges for the scale-up and production of stable raw materials. Thus, developing drug preparations is difficult, despite the empirical evidence regarding the bioactivity of acetogenin compounds. Moreover, high doses of acetogenins can be neurotoxic and may cause neurodegenerative disorders. Some alkaloids present in *A. muricata* are also believed to affect nerve cells. However, research on the neurotoxicity of annonacin states that neurodegenerative conditions caused by these compounds arise due to continuous exposure or consumption. Further research on the toxicity of *A. muricata* and clinical trials testing the pure compounds are needed to fully elucidate its pharmacological activities and ensure the safety of *A. muricata* as a potential drug for various diseases.

## Figures and Tables

**Figure 1 molecules-27-01201-f001:**
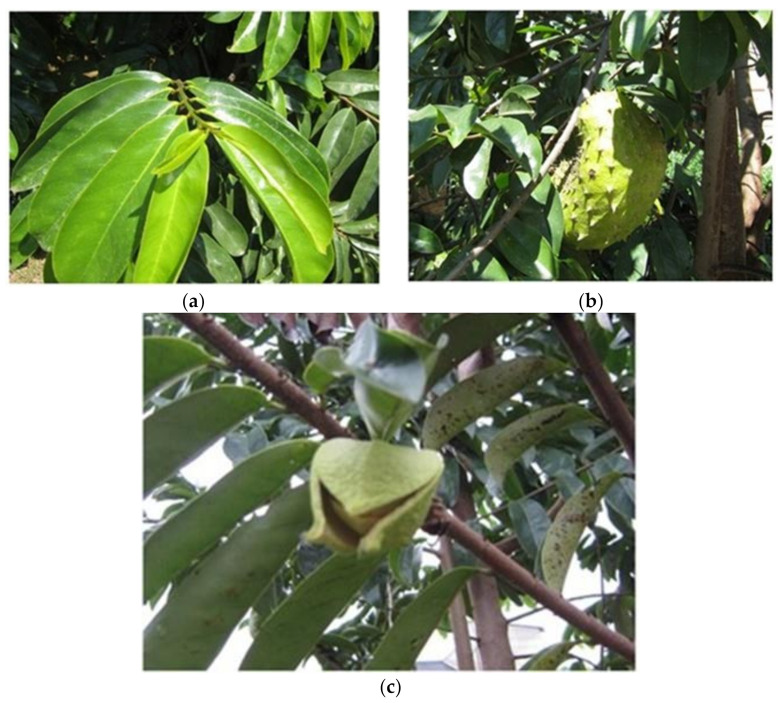
(**a**) Leaves of *A. muricata* with an obovate, oblate, and acuminate shape. The leaf surface is dark green, with a thick and glossy upper surface. (**b**) The fruits are dark green and prickly. (**c**) The flower petals are thick and yellowish. The outer petals meet at the edges without overlapping and are broadly ovate, tapering to a point with a heart-shaped base. The inner petals are oval-shaped and overlap.

**Figure 2 molecules-27-01201-f002:**
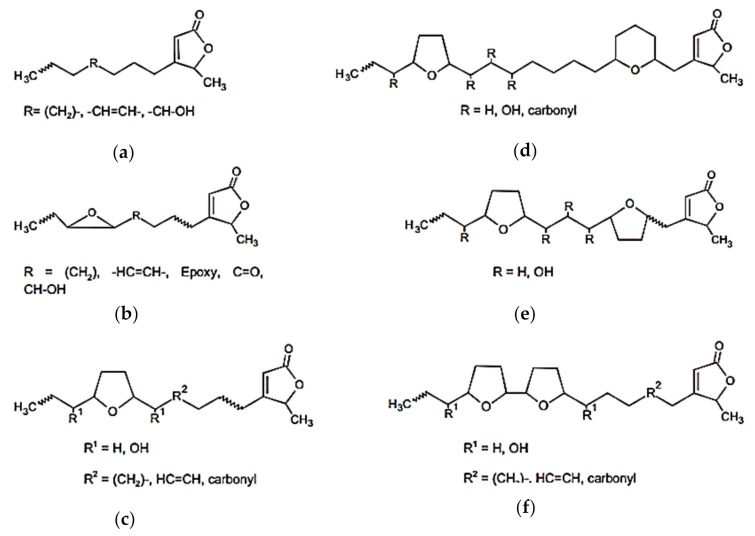
Acetogenin compounds in *A. muricata*. (**a**) Linear structure, (**b**) epoxy acetogenin, (**c**) mono THF, (**d**) mono tetrahydrofuran, mono tetrahydropyran acetogenin, (**e**) bis THF-nonadjacent acetogenin, (**f**) and bis THF-adjacent acetogenin [4].

**Figure 3 molecules-27-01201-f003:**
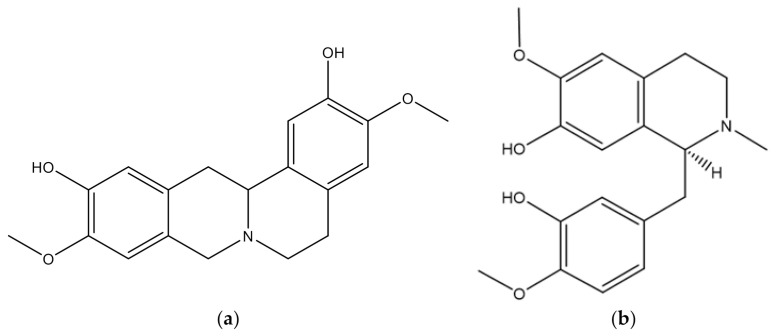
The most abundant alkaloids in *A. muricata*: (**a**) coreximine and (**b**) reticuline.

**Figure 4 molecules-27-01201-f004:**
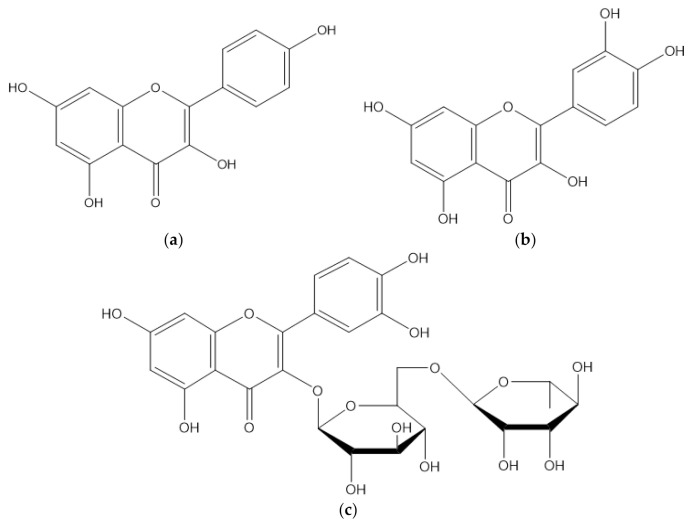
The most abundant flavonoid in *A. muricata*: (**a**) kaempferol (**b**) quercetin, and (**c**) rutin.

**Figure 5 molecules-27-01201-f005:**
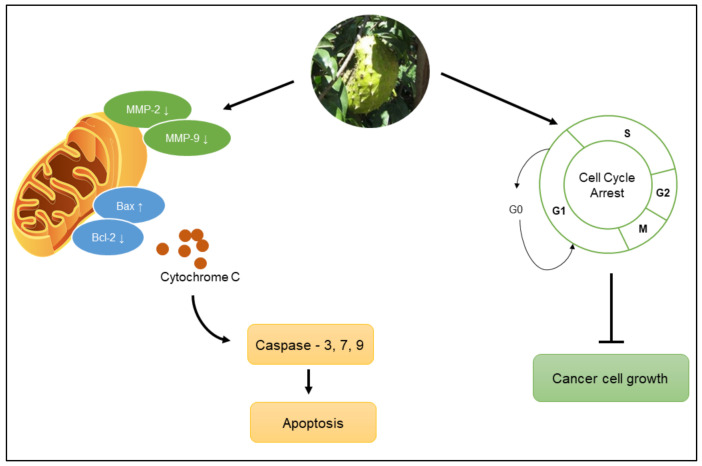
Anticancer mechanism of *A. muricata*.

**Figure 6 molecules-27-01201-f006:**
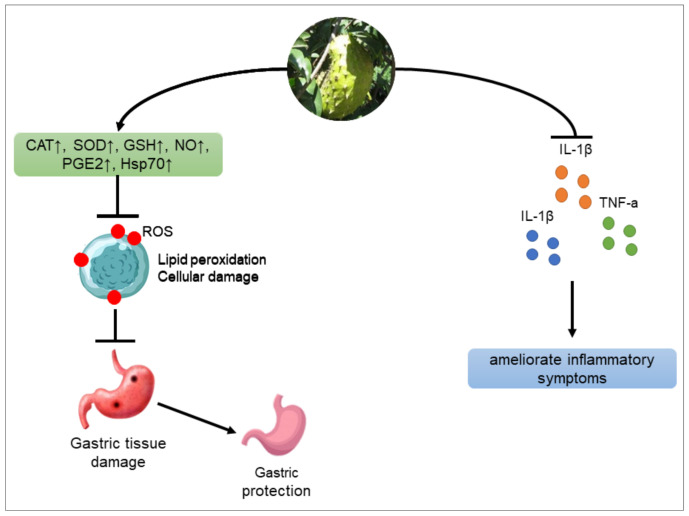
Antiulcer mechanisms of *A. muricata*.

**Figure 7 molecules-27-01201-f007:**
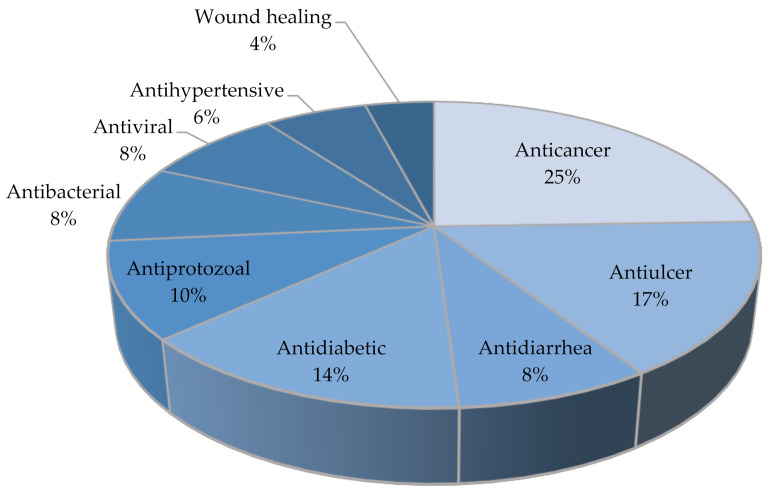
Distribution of pharmacological activities of *A. muricata*.

**Figure 8 molecules-27-01201-f008:**
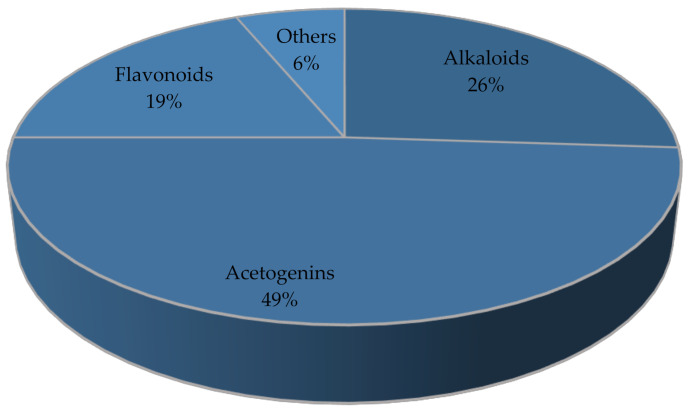
Proportions of phytochemical compounds in *A. muricata*.

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
