# Peer review of "Pharmacological Activities of Soursop (Annona muricata Lin.)"

_molecules, 2022, doi:10.3390/molecules27041201_

Round 1

Reviewer 1 Report

The manuscript entitled "Pharmacological activity of soursop (Annona muricata Lin.)" summarize the traditional uses of soursop as well as the serveral bioactivities already demonstrated. The field of the study is not novel and there are some reviews already published within the last years summarizing the traditional uses and biological activities of soursop. Most of these studies have been focused on one group of bioactive compounds (acetogenin, phenolic compounds, ...). However, the present study compile information about the different bioactive compounds and bioactivities. The manuscript is generally well written. However, some comments can help to improve the scientific quality of the manuscript.

Pharmacological activities were described according to different mechanisms (antioxidant and antiinflamatory) and targeted diseases (anticancer, antidiabetes...). In my opinion, the biologic mechanisms must be appear isolated from the target diseases. I mean, a specific extract or bioactive compound can combat cancer or diabetes through the inflammation/antioxidant activity. Indeed, authors divided section 5.1. Anticancer and 5.2. Antioxidants and the 5.2 section starts saying: “Antioxidants are useful to fight cancer by inhibiting the carcinogenesis processes [61].” Line 167. I herein propose to start explaining the pharmacological activities in a general overview of main mechanisms of action to them specify the main achievements for targeted diseases. Furthermore, a lack of general criticism was found. What are the main weakness in the evidences already found? Is there any gap in the literature? The methods are standardized? What about bioavailability and pharmacokinetics? The studies already performed were in vitro, ex vivo, or in vivo? Are there any correlation between the body of the research?

Author Response

The manuscript entitled "Pharmacological activity of soursop (Annona muricata Lin.)" summarize the traditional uses of soursop as well as the several bioactivities already demonstrated. The field of the study is not novel and there are some reviews already published within the last years summarizing the traditional uses and biological activities of soursop. Most of these studies have been focused on one group of bioactive compounds (acetogenin, phenolic compounds, ...). However, the present study compile information about the different bioactive compounds and bioactivities. The manuscript is generally well written. However, some comments can help to improve the scientific quality of the manuscript.

Answ.:

We are very grateful for reviewer 1 advices that helps us improve the quality of our writing.

Pharmacological activities were described according to different mechanisms (antioxidant and antiinflamatory) and targeted diseases (anticancer, antidiabetes...). In my opinion, the biologic mechanisms must be appear isolated from the target diseases. I mean, a specific extract or bioactive compound can combat cancer or diabetes through the inflammation/antioxidant activity. Indeed, authors divided section 5.1. Anticancer and 5.2. Antioxidants and the 5.2 section starts saying: “Antioxidants are useful to fight cancer by inhibiting the carcinogenesis processes [61].” Line 167. I herein propose to start explaining the pharmacological activities in a general overview of main mechanisms of action to them specify the main achievements for targeted diseases.

Answ.:

Yes, we absolutely agree with reviewer 1 suggest about antioxidant and anti-inflammatory is one of the process of our immunity system against another disease.

So we changed the subtitle to be more disease targeted start on page 10 line 141, according to the reviewer's suggestion.

Furthermore, a lack of general criticism was found. What are the main weakness in the evidences already found? Is there any gap in the literature? The methods are standardized? What about bioavailability and pharmacokinetics? The studies already performed were in vitro, ex vivo, or in vivo? Are there any correlation between the body of the research?

Answ.:

Thank you for your suggestions, we aim to analyze the references that we get to study the development of research on A. muricata because even though it is just starting, we are also involved in it. We include references with high criteria and reliable data. However, so far we have not found any information regarding the bioavailability and pharmacokinetics of the extract or single compound of A. muricata. Most of the reference that we found are in vitro studies, and some in vivo which state the toxicity of extracts of A. muricata. From the results of this study, it is hoped that researchers can lead more precisely towards the progress of testing towards the commercialization of A. muricata both as herbal medicines and drugs with a single compound of A. muricata. We state our critics that added in last paragraph in conclusion section.

Reviewer 2 Report

The submitted manuscript describes and discusses the original description carried out to provide an overview of the botanical description, traditional uses, compounds, and pharmacological activity of A. muricata by summarizing the existing literature.The manuscript presents information that are expected to be of large interest for the scientific community. It is an interesting study with an interesting approach. The paper in the whole is well designed and results sound. Nevertheless, the manuscript needs a major revision:

  • In the introduction part should be more highlighted the main aim of the paper, and additionally, what is the novelty of carried research work.
  • How do the Authors select the analytes? The rational of the choice of the selected biologically active compounds studied is missing and should be clearly discussed.
  • Quality of the figures must be improved.
  • Concluding remarks: Information given in Conclusion is not very useful. It would be more useful without the general facts.

Author Response

The submitted manuscript describes and discusses the original description carried out to provide an overview of the botanical description, traditional uses, compounds, and pharmacological activity of A. muricata by summarizing the existing literature. The manuscript presents information that are expected to be of large interest for the scientific community. It is an interesting study with an interesting approach. The paper in the whole is well designed and results sound. Nevertheless, the manuscript needs a major revision:

In the introduction part should be more highlighted the main aim of the paper, and additionally, what is the novelty of carried research work.

Answ.:

Thank you for reviewer 2 advice, as we wrote on the manuscript, we provide an overview of the botanical description, traditional uses, compounds, and pharmacological activities of A. muricata to know the development of research on A. muricata in order to be able to direct the direction of research.

How do the Authors select the analytes? The rational of the choice of the selected biologically active compounds studied is missing and should be clearly discussed.

Answ.:

It is hard to understand what the reviewer 2 meant. Can reviewer 2 explain the details in question?

If the reviewer mean is about phytochemical properties, we added alkaloids in the table 2.

Quality of the figures must be improved.

Answ.:

Yes, indeed, thank you for remind us.

To get a better picture on figures 3 and 4, we use ChemDraw software ver. 16. 0.1.4.

Concluding remarks: Information given in Conclusion is not very useful. It would be more useful without the general facts.

Answ.:

Again, it is hard to understand what the reviewer 2 meant.

We added several sentences in conclusion section. We are very grateful if reviewer 2 would like explain us in more detail what is meant in order to make our writing even better.

Reviewer 3 Report

The manuscript written by Mutakin and colleagues is very interesting, once it envisages the pharmacological and toxic properties of this plant. In my opinion it is well written, presents great tables, but it lacks a mechanism of action figure comprising the main pharmacological activities. Moreover, in the abstract, authors affirm: “A. muricata is 13 used as an anticancer (23%), antioxidant (12%), antiulcer (11%), antidiabetic (9%), antihypertensive 14 (5%), and anti-inflammatory agent (11%), as a skin disease treatment, and to reduce fever and treat 15 diarrhea (3%)” . What is the origin of those percentages? Figure 3: please consider alkaloidS Figure 4: please consider compound instead of group I believe after these considerations the manuscript will be ready to be accepted for publication.

Author Response

The manuscript written by Mutakin and colleagues is very interesting, once it envisages the pharmacological and toxic properties of this plant. In my opinion it is well written, presents great tables, but it lacks a mechanism of action figure comprising the main pharmacological activities.

Answ.:

We are very grateful for your advice, we improved our manuscript by compiled two figures of the mechanism of action of the active compound of A. muricata as an anticancer and antiulcer as Figure 5 and 6, which has been clearly proven from previous papers.

Moreover, in the abstract, authors affirm: “A. muricata is 13 used as an anticancer (23%), antioxidant (12%), antiulcer (11%), antidiabetic (9%), antihypertensive 14 (5%), and anti-inflammatory agent (11%), as a skin disease treatment, and to reduce fever and treat 15 diarrhea (3%)”. What is the origin of those percentages?

Answ.:

Thank you for reminding us to indicate the number of manuscripts we refer to so that we can clearly show where the numbers are coming from. We added description of the total paper that is the source of reference for the pharmacological activities and phytochemical properties of our manuscript.

Figure 3: please consider alkaloids Figure 4: please consider compound instead of group I believe after these considerations the manuscript will be ready to be accepted for publication.

Answ.:

Once again, we are very grateful for reviewer 3 advices.

Please find alkaloids name list in Table 2 and improved Figure 3 and 4 qualities.

After this improvement, we also hope that our manuscript is ready to be accepted for publication.

Round 2

Reviewer 3 Report

The authors have restructured the article and in my opinion it is better stated now.  However, some points still need an adjustment or an explanation, as it follows:  

  • In the abstract, please consider: "From 49 research articles that were obtained from 1981-2021, the pharmacological properties described for A. muricata were anticancer (24%), antiulcer (18%), antidiabetic (14%), antiprotozoal (10%), antidiarrheal (8%), antibacterial (8%), antiviral (8%), antihypertensive (6%) and wound healing (4%)." agent; as a skin disease treatment; and to reduce fever and treat diarrhea (8%) was reported during 1981–2021.  The pharmacological properties must be in accordance with the conclusion section and figure 7. 
  • Line 145: "cell cycle arrest". Please, insert it. 
  • Line 161: "Another mechanism related to the anticancer activity of A. muricata is modulation of antioxidant enzyme activities. A study reported that 50% ethanol extract of A. muricata leaves led to upregulation of the expression of the antioxidant enzyme superoxide dismutase-1 (SOD1), which catalyzes the breakdown of superoxide into oxygen (O2) and hydrogen peroxide (H2O2), preventing cellular damage [73](Figure 5)." Figure 5 does not contain this mechanism of action.
  • Figure 5: the right part of this figure is not correct. A muricata inhibits cell cycle progression (not arrest), which, in turn, blocks cell proliferation.  The other signal of inhibition with the data "necrosis, cytotoxicity and proliferation" must be revised. Moreover, the text does not bring an explanation about necrosis... please insert the reference for it.  

Author Response

Thank you for review the above manuscript.

Answers to comments from Reviewer #3

The authors have restructured the article and in my opinion it is better stated now.  However, some points still need an adjustment or an explanation, as it follows: 

In the abstract, please consider: "From 49 research articles that were obtained from 1981-2021, the pharmacological properties described for A. muricata were anticancer (24%), antiulcer (18%), antidiabetic (14%), antiprotozoal (10%), antidiarrheal (8%), antibacterial (8%), antiviral (8%), antihypertensive (6%) and wound healing (4%)." agent; as a skin disease treatment; and to reduce fever and treat diarrhea (8%) was reported during 1981–2021.  The pharmacological properties must be in accordance with the conclusion section and figure 7.

Line 145: "cell cycle arrest". Please, insert it.

Line 161: "Another mechanism related to the anticancer activity of A. muricata is modulation of antioxidant enzyme activities. A study reported that 50% ethanol extract of A. muricata leaves led to upregulation of the expression of the antioxidant enzyme superoxide dismutase-1 (SOD1), which catalyzes the breakdown of superoxide into oxygen (O2) and hydrogen peroxide (H2O2), preventing cellular damage [73](Figure 5)." Figure 5 does not contain this mechanism of action.

Answ.:

We are very grateful for reviewer’s advices that helps us improve the quality of our writing.

After we check the abstract and have some discussion, thanks you for the reviewer’s correction, we agreed with reviewer to remove the sentence after the quotation marks, ie; agent; as a skin disease treatment; and to reduce fever and treat diarrhea (8%) was reported during 1981–2021.

Please check line 145 becomes line 153 in new draft, we adjust according to the reviewer's suggestions.

Please also check line 161 that becomes to line 174 in new draft, thanks for the reviewer's correction, we adjust according to the reviewer's suggestions.

Figure 5: the right part of this figure is not correct. A muricata inhibits cell cycle progression (not arrest), which, in turn, blocks cell proliferation.  The other signal of inhibition with the data "necrosis, cytotoxicity and proliferation" must be revised. Moreover, the text does not bring an explanation about necrosis... please insert the reference for it. 

Answ.:

Thank you for responding and correcting the figures we added to the last draft, we really appreciate your suggestions. We checked once again, according to the literature that the figure we show is a cell cycle arrest chart or checkpoint, if there is DNA damage in the cell, the cell will evaluate and stop the cell growth process which will automatically inhibit the replication of cancer cells. So we changed it as shown in figure 5, page 11 of the new draft.
